# Posting patterns in peer online support forums and their associations with emotions and mood in bipolar disorder: Exploratory analysis

**Glorianna Jagfeld**[1]*, **Fiona Lobban**[1], **Robert Davies**[2], **Ryan L. Boyd**[2,3,4,5,6], **Paul Rayson**[7], **Steven Jones**[1]

1 Spectrum Centre for Mental Health Research, Division of Health Research, Lancaster University, Lancaster, United Kingdom, 2 Department of Psychology, Lancaster University, Lancaster, United Kingdom, 3 Security Lancaster, Lancaster University, Lancaster, United Kingdom, 4 Data Science Institute, Lancaster University, Lancaster, United Kingdom, 5 Obelus Institute, Behavioral Science Division, Washington D.C., United States of America, 6 Department of Computer Science, Stony Brook University, Stony Brook, NY, United States of America, 7 UCREL Research Centre, School of Computing and Communications, Lancaster University, Lancaster, United Kingdom

* g.jagfeld@lancaster.ac.uk

**Data Availability Statement:** There are ethical restrictions which prevent the public sharing of data utilized in the course of this study. Data are available upon request from Lancaster University

## Abstract

### Background

Mental health (MH) peer online forums offer robust support where internet access is common, but healthcare is not, e.g., in countries with under-resourced MH support, rural areas, and during pandemics. Despite their widespread use, little is known about who posts in such forums, and in what mood states. The discussion platform Reddit is ideally suited to study this as it hosts forums (subreddits) for MH and non-MH topics. In bipolar disorder (BD), where extreme mood states are core defining features, mood influences are particularly relevant.

### Objectives

This exploratory study investigated posting patterns of Reddit users with a self-reported BD diagnosis and the associations between posting and emotions, specifically: 1) What proportion of the identified users posts in MH versus non-MH subreddits? 2) What differences exist in the emotions that they express in MH or non-MH subreddit posts? 3) How does mood differ between those users who post in MH subreddits compared to those who only post in non-MH subreddits?

### Methods

Reddit users were automatically identified via self-reported BD diagnosis statements and all their 2005–2019 posts were downloaded. First, the percentages of users who posted only in MH (non-MH) subreddits were calculated. Second, affective vocabulary use was compared in MH versus non-MH subreddits by measuring the frequency of words associated with

([https://doi.org/10.17635/lancaster/researchdata/589](https://doi.org/10.17635/lancaster/researchdata/589)) for researchers who meet the criteria for access to confidential data by signing a data usage agreement.

**Funding:** This study was completed as part of a PhD studentship for GJ, funded by the Faculty of Health and Medicine at Lancaster University, UK ([https://www.lancaster.ac.uk/health-and-medicine/](https://www.lancaster.ac.uk/health-and-medicine/), internal cost code: HRA6043). The funders had no role in study design, data collection and analysis, decision to publish, or preparation of the manuscript.

**Competing interests:** The authors have declared that no competing interests exist.

positive emotions, anxiety, sadness, anger, and first-person singular pronouns via the LIWC text analysis tool. Third, a logistic regression distinguished users who did versus did not post in MH subreddits, using the same LIWC variables (measured from users' non-MH subreddit posts) as predictors, controlling for age, gender, active days, and mean posts/day.

## Results

1) Two thirds of the identified 19,685 users with a self-reported BD diagnosis posted in both MH and non-MH subreddits. 2) Users who posted in both MH and non-MH subreddits exhibited less positive emotion but more anxiety and sadness and used more first-person singular pronouns in their MH subreddit posts. 3) Feminine gender, higher positive emotion, anxiety, and sadness were significantly associated with posting in MH subreddits.

## Conclusions

Many Reddit users who disclose a BD diagnosis use a single account to discuss MH and other concerns. Future work should determine whether users exhibit more anxiety and sadness in their MH subreddit posts because they more readily post in MH subreddits when experiencing lower mood or because they feel more able to express negative emotions in these spaces. MH forums may reflect the views of people who experience more extreme mood (outside of MH subreddits) compared to people who do not post in MH subreddits. These findings can be useful for MH professionals to discuss online forums with their clients. For example, they may caution them that forums may underrepresent people living well with BD.

## Introduction

Experiences of 'Manic, Mixed, or Hypomanic episodes or symptoms' alternating with 'Depressive episodes or periods of depressive symptoms' [1] that interfere with functioning, constitute core criteria for a bipolar disorder (BD) diagnosis [1, 2]. Prevalence rate estimates for bipolar spectrum disorders range from 0.1% (India), 2% (England), to 4.4% (US) across 22 different European, American, and Asian countries [3–5]. The characteristics and outcomes of people meeting BD criteria are diverse, with some functioning on a high level [6–9], achieving long-term remission [10] and living well [11–13]. However, many individuals with a BD diagnosis experience recurring mood episodes [14, 15] and require life-long treatment [16]. Moreover, BD has the highest rate of suicide across all mental health (MH) diagnoses [16, 17]. Therefore, it is important to understand how people living with BD can best be supported to live well.

The internet, including social media and online support forums, plays an increasingly important role in sharing information and support for health and MH concerns [18, 19]. Like the general population, most people with a (BD) diagnosis are internet users [20, 21] and seek information about BD online [22]. In 2014–15, 13%-17% of people with a BD, major depressive disorder, or psychotic disorder diagnosis reported posting and seeking information on their condition in online support forums or social media [23, 24]. Since then, the importance of online MH forums has likely increased. The online discussion platform Reddit ([reddit.com](reddit.com)) is one of the most visited internet sites worldwide and hosts subforums (subreddits) for a variety of topics, ranging from general interests to specific topics, including MH [25]. In 2019, the number of members in BD subreddits (bipolar, BipolarReddit, bipolar2, bipolarSOs) increased

by more than 50%. In light of the global Covid-19 pandemic, where in-person care has become more difficult to access for many people, international BD experts have highlighted the importance of online MH support and the "great need" to "train and support clinicians to go where people are already online" [26]. People may also use online forums in addition to in-person MH care since quantitative [27, 28] and qualitative [29, 30] evidence shows that online anonymity affords personal self-disclosures and discussions of sensitive and stigmatised issues.

There are a number of content analyses of BD online forums [31–37], including subreddits [38]. Jagfeld, Lobban, Rayson, et al. [39] characterised the demographics (age, gender identity, country of origin, comorbidities) of Reddit users with a self-reported BD diagnosis. However, little is known about the users of online MH support forums and why some users chose to engage in them and others not. This knowledge is both relevant to improve online MH support forum design and to contextualise the increasing body of research relating to such forums. Because users may post in MH-focused as well as other subreddits, Reddit is uniquely useful to study who chooses to engage in online MH support forums [40]. However, only Reddit users who post both in MH and non-MH subreddits can be considered for such within-person comparisons. Ireland and Iserman [40] noted that only one-third of Reddit users who had posted at least 50 words in anxiety subreddits had also posted at least 50 words in other subreddits; no similar studies exist for other MH concerns. Therefore, the first research question investigates what proportion of Reddit users with a self-reported BD diagnosis post in MH and non-MH focused subreddits using the same account.

Research using NLP methods with online posts of people with a BD diagnosis has mainly focused on language differences between users with a BD and other or no MH diagnosis and used these to build machine-learning classifiers to identify social media users likely experiencing BD symptoms [41]. While earlier studies compared all of users' content [42, 43], more recent work excluded MH-related posts [44–46] aiming to build classification systems that could also detect users with MH issues who do not discuss MH concerns online. Yet, to date, only one study has investigated differences between posts in MH and non-MH contexts of the *same* users. Ireland and Iserman [40] found that people used more anxious language in anxiety-focused subreddits compared to their posts in other subreddits. Such a comparison could provide insights into the extent to which findings from earlier studies that compared all of users' posts were due to language differences between people with MH and without MH issues or rather due to the different contexts of the posts in MH vs. non-MH subreddits that may impact on their emotional content. However, no comparable analyses exist for mood disorders or BD. Hence, the second research question examines what differences exist in the emotions that Reddit users with a BD diagnosis express in MH and non-MH subreddit posts.

Extreme and changing mood states are the core defining features of BD [1, 2]. Yet, very little research has been conducted so far on the relationship between mood and engaging with online MH forums for people with a BD diagnosis. Gathering data on peoples' emotions and mood is difficult. Surveys require people to fit their complex emotional experiences into rigid scales or describe them with a fixed set of preselected terms. Asking about past emotions and mood might suffer from recall bias [47, 48]. In general, any kind of self-report method may be biased by social desirability [49–51]. Moreover, in online forums, users may not be active anymore and therefore cannot be invited to active study participation.

Computational text analysis approaches such as Linguistic Inquiry and Word Count (LIWC) [52] can provide non-reactive behavioural indicators of emotions and mood that sidestep a lot of the problems with self-reported mood measures. LIWC provides indicators of peoples' affective and cognitive processes based on what percentage of the words in their language falls into categories such as positive emotion, sadness, and anxiety. A number of studies have applied LIWC to online posts of people with MH issues, including BD [42–45, 53, 54]. Park

and Conway [54] provide evidence using LIWC that Reddit users' negative emotions decrease, and positive emotions increase with every post they make in a depression support subreddit. Text emotion analysis with machine learning methods [55] has shown that users' emotional tone improves during exchanges in four large MH subreddits for depression, suicidality, BD, and anxiety when comparing their initial submission to their last comment in the same thread. As these are the only two studies looking at the relationship of emotions and mood and posting in online MH communities, the third research question addresses a critical gap in answering how mood differs between Reddit users with a BD diagnosis who post in MH subreddits and those who only post in non-MH subreddits.

## Aim, research questions, and contributions

To summarise, this study aims to shed more light on the posting patterns in online support forums and their associations with the emotions and mood of people with a BD diagnosis. By applying psychological text analysis methods to the public online posts of a large sample of Reddit users with a self-reported BD diagnosis this exploratory study investigates the following research questions:

RQ 1: What proportion of Reddit users with a BD diagnosis posts in MH and non-MH subreddits?

RQ 2: What differences exist in the emotions that Reddit users with a BD diagnosis express in MH and non-MH subreddit posts?

RQ 3: How does mood differ between Reddit users with a BD diagnosis who post in MH subreddits and those who only post in non-MH subreddits?

This study makes the following contributions to MH research:

- Provides the first quantitative evidence on the posting behaviour of people with a BD diagnosis and how demographic and emotional factors may impact posting in MH forums specifically (see Fig 1 for a visual summary of the results)

- Demonstrates differences in the emotional content of MH vs non-MH subreddit posts

- Shows how to use established NLP methods to provide quantitative data and generate findings relevant for MH researchers and professionals

- Provides a large dataset of 21M Reddit posts by 20K Reddit users with a self-reported BD diagnosis that is available for future research

## Methods

Reporting of this study follows the STROBE guidelines for cross-sectional studies [56].

## Reddit user identification and inclusion and exclusion criteria

Reddit users who stated on the platform that they had received a BD diagnosis from a professional were automatically identified via self-reported diagnosis statements like 'I was diagnosed with bipolar disorder' [see 39]. All posts of candidate users (id, text, timestamp created at, subreddit) between 01/2005 (site inception) and 03/2019 (data available at time of data collection in 05/2019) were downloaded via the RedditAPI praw [57]. Candidate users were filtered by matching all of their posts against 74 exclusion patterns like 'not officially diagnosed' or 'self-diagnosed'. All posts of these users were also matched against self-reported diagnosis

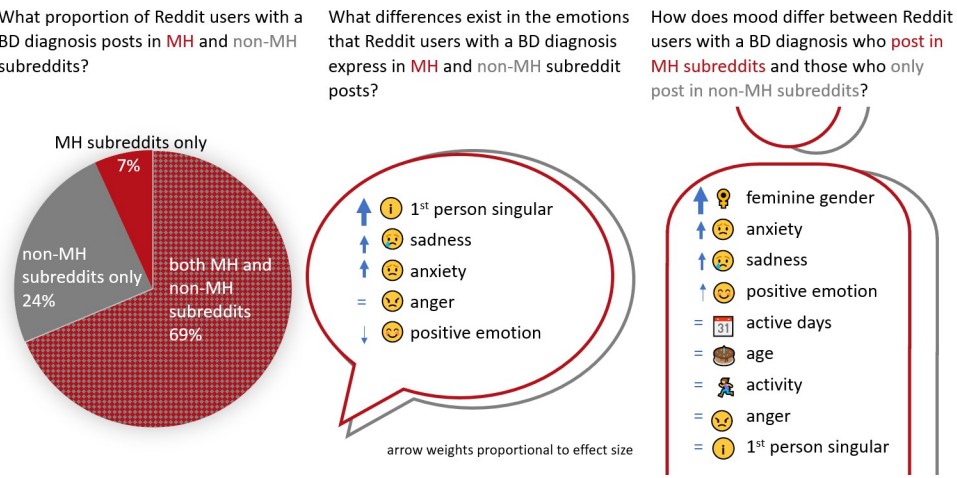

**Fig 1. Visual summary of the results.**

statements for nine other MH diagnoses [39]. A major depressive disorder or psychotic disorder diagnosis constitutes an exclusion criterion for BD according to DSM-5 [2]. However, around 40% of people who are diagnosed with BD first received a major depressive disorder diagnosis [58, 59]. Therefore, only users who also reported a psychotic disorder diagnosis were removed. The Self-reported BD diagnosis (S-BiDD) dataset comprisesall posts of the remaining users.

RQ2 only concerned a subset of users with at least four posts of at least 25 words each in both MH and non-MH subreddits. Comparable studies of Reddit language using LIWC required at least four posts in depression subreddits [54] or five posts overall [60]. To address RQ3, these users who had posted in MH subreddits were compared against users who had no posts in MH subreddits and at least eight posts with at least 25 words each in non-MH subreddits. Users for which the NLP methods could not assign the control variables age or gender (see Section 'Statistical analyses' below) were excluded in the analyses to address RQ3.

## MH subreddit identification

To determine whether posts were posted in subreddits with a MH focus, subreddit-to-topic assignments were amended and extended from existing resources as there is no official subreddit topic taxonomy from Reddit (see S1 Appendix for method details). The final four-level taxonomy [61] comprises 30,867 topic-categorised subreddits, including 158 MH-related subreddits, of which 37 are bipolar-specific. Only 3.6% of the posts by people with a self-reported BD diagnosis are in a subreddit not assigned to a topic. See S1 Table for an overview of the top 10 BD, MH (not BD-specific) and non-MH subreddits with most posts in the S-BiDD dataset.

## Language analysis

LIWC [52] was used to obtain text-analytic measures of peoples' emotions and mood. In this study, users' emotions–usually short-lived (seconds [62]), high in intensity and in response to the evaluation of a stimulus [63]–are operationalised via the LIWC scores of single posts. Averaged LIWC scores of multiple posts over a longer timespan are assumed to represent users' mood, which is of longer duration, lower intensity and not in direct temporal relationship to a stimulus [63].

Recent NLP research focuses on emotion text analysis systems based on machine learning or deep learning methods [e.g., 64–66]. Yet, LIWC was considered the most suitable emotion analysis tool for this study for four main reasons. First, this type of language analysis via expert-created dictionaries is rooted in over 100 years of empirical research in the behavioural sciences and there is a large evidence base for the psychometric validity of this approach [e.g., 67–71]. Particularly for the LIWC emotion categories, Kahn et al. [72] demonstrated that the scores for positive emotion, negative emotion and sadness significantly differ in the expected ways in peoples' responses to discrete stimuli. The participants had higher positive emotion and lower negative emotion and sadness scores when writing about an amusing versus sad life experience and describing their feelings after exposure to a funny versus sad video clip. Newell et al. [73] showed that more negative emotion, less positive emotion, and a higher use of first-person singular pronouns correlates with peoples' subjectively reported stress level and their blood pressure. Moreover, LIWC emotion variables also correlate with established daily and seasonal mood fluctuations within individuals and for entire populations [74–76]. In contrast to this, machine learning-based NLP methods require labelled training data and usually only perform well on data that is similar to the training data. While Google recently released an emotion-labelled dataset of Reddit posts [65], it is not specific to people with MH issues and therefore not suitable to train or fine-tune an emotion classifier for the aims of this study. Second, LIWC emerged from Pennebaker's work on narrative therapy [e.g., 77], which is widely recognised amongst clinicians, while most of them are not familiar with machine learning and deep learning approaches. Applied psycholinguistic research using LIWC is very active as evidenced by many studies published just in the past two years [e.g., 78–81]. Third, LIWC yields results that are easy to interpret because they directly relate to the frequencies of words in the dictionaries. This is not the case for machine learning or deep learning systems whose predictions are often opaque and hard to trace back to the input [82]. Fourth, several previous studies have applied LIWC to online posts of people with MH issues, including BD [40, 42–45, 53, 54, 83], to the extent that most research on BD using NLP methods has used LIWC [41]. This allows for a straightforward comparison of the present results with previous work in the discussion or RQ2 and RQ 3.

Five LIWC categories were preselected for the analyses (see Table 1), including all affective processes subcategories (*positive emotion*, *anxiety*, *anger*, *sadness*). A number of previous studies used these categories to predict perceived negative/positive emotions from text [see 54]. Additionally, the linguistic variable *1st pers. sg.*, measuring the frequency of first person singular pronouns, was included because previous research pervasively links it to mental distress [84]. Only posts with at least 25 words were included in the analyses because LIWC estimates for shorter texts are less reliable [60, 85].

**Table 1. LIWC variables related to emotions and mental distress selected for this study.**

| LIWC variable | Example vocabulary [52, Table 1] |
| --- | --- |
| **Affective processes** | |
| • Positive emotion (posemo) | love, nice, sweet |
| • Negative emotion | |
| ○ Anxiety | worried, fearful |
| ○ Anger | hate, kill, annoyed |
| ○ Sadness | crying, grief, sad |
| **Linguistic variables** | |
| First person singular (1st pers. sg.) | I, me, mine |

## Statistical analyses

To address RQ1, the percentages of users who posted only in MH subreddits, only in non-MH subreddits or in both were calculated. Previous work considered only Reddit users with at least four posts in MH subreddits as active contributors and those with fewer posts as *lurkers* [54]. Therefore, the number of users with at least four posts in both MH and non-MH subreddits was also calculated, and, for comparison, the number of users with at least eight posts who posted only in MH or non-MH subreddits.

To address RQ 2, the mean LIWC variable scores of the posts of users in MH subreddits were compared to the mean scores in non-MH subreddits via dependent t-tests with Cohen's d as the effect size measure [86].

To address RQ3, a logistic regression model was fitted via the R [87, Windows version 4.1.0] glm function to predict whether a user had posted in MH subreddits based on the mean LIWC variable scores of their non-MH subreddit posts and the control variables described below. Model fit for the model including only the control variables versus the model including control and LIWC variables was compared using a likelihood ratio test, following best practice guidance for linear mixed-effects models in psychological science [88]. See S2 Appendix for method details. The outcome variable *posted in MH* was coded as 1 for users who had posted in MH subreddits and 0 for those who had not (see 'Reddit user identification and inclusion and exclusion criteria'). Importantly, only non-MH subreddit posts were used to measure mood to provide a more comparable source of measures for users from both groups. LIWC emotion variables might be affected by the communicative context [89]. Only including posts made in non-MH subreddits avoided conflating mood and context.

The model also included the four control variables, age, gender, active days, and activity. *Age* was users' average age, considering all their posts in the S-BiDD dataset, as automatically inferred from their post texts via natural language processing (NLP) models [see 39]. There is some evidence for a decrease in perceived stigma, perceived discrimination, and stigmatising newspaper articles for MH issues in recent years [90–92]. Moreover, younger cohorts use social media differently than older adults. For example, 68% of 18–35 year old young adults with MH issues but only 54% of 36–65 year olds reported sharing personal experiences about MH issues as the main reason for their social media use [93]. Therefore, it appeared relevant to check whether younger users would be more likely to post in MH subreddits. *Gender* corresponded to users' binary gender as automatically inferred from their gender identity via self-reported statements like '*I'm 34M*' or their performed gender from the username or post texts via NLP models [see 39] (0 = feminine, 1 = masculine). Gender was included as a control variable because previous research found men were less likely to disclose and seek support for MH issues than women [94–97], although it is unknown how this plays out online. Finally, *active days* corresponded to the number of days between user's first and last post in the S-BiDD dataset and *activity* to the number of posts divided by *active days*. These two variables were included to check if users' overall activity on Reddit had an impact on their likelihood to post in MH support forums. For example, someone who is more active on Reddit generally and has been for a longer time may be more likely to have come across and post in a wider range of subreddits, including MH related ones.

## Ethical considerations and transparency

The study is consistent with the Helsinki Declaration [98], the ethics guidelines for internet-mediated research of the British Psychological Society [99] and received approval from the Lancaster University Faculty of Health and Medicine Research Ethics Committee. Current ethical guidelines permit the waiving of explicit consent of internet users if only public content is

analysed without a threat to anonymity or confidentiality [e.g., 99]. In line with this, the present paper only shares aggregated results of anonymous Reddit users. See [39] for further ethical considerations in collecting the S-BiDD dataset. The purpose of this work is to better understand online users with a self-reported BD diagnosis and only comparisons within this group are conducted. Therefore, the methods and results presented here do not serve to identify online users with MH issues. People with lived experience of BD and using social media that were consulted did not raise any ethical concerns of this research (see Section Feedback from people with lived experience).

The S-BiDD dataset is available for non-commercial research [100] upon request via rdm@lancaster.ac.uk and under a data usage agreement that specifies ethical usage as detailed in [39]. The code is publicly available [101].

## Involvement of people with lived experience

In preparation for this study, four volunteers with lived experience of BD and using online support forums recruited via PeopleInResearch were invited to share their relevant experiences and comment on the research plan in individual online meetings which lasted 30–60 minutes. The volunteers had a positive attitude towards the study as they were generally not aware of other work analysing online forum posts of people with a BD diagnosis. They agreed that analysis of public online posts was justifiable if users' anonymity was safeguarded. Most of the volunteers reported that they were posting in MH support forums when in low (but not very low) mood. Moreover, they all gradually reduced their engagement with online forums once they were better able to manage their BD and found a job or other projects they engaged with which may have been accompanied with more positive mood. Since sadness and anxiety were found to constitute core symptoms of low mood [102], this led to the following expectations: RQ 2: MH subreddit posts express more anxiety and sadness; RQ3: Users who post in MH subreddits have higher long-term measurements of anxiety and sadness. After completion of the analyses, the volunteers were contacted again to provide feedback on our interpretations of the results in individual online meetings.

## Results

### Results summary

The final Self-reported BD diagnosis (S-BiDD) dataset comprises 21,407,595 posts (both submissions (thread starts) and comments) of 19,685 users with a self-reported BD diagnosis. Fig 1 provides a visual summary of all results. Two thirds of the identified 19,685 users with a self-reported BD diagnosis posted in both MH and non-MH subreddits, while 24% never posted in MH subreddits. Users who posted in both MH and non-MH subreddits used more first-person singular pronouns and exhibited more sadness and anxiety, but less positive emotion in their MH subreddit posts. Feminine gender and expressing more anxiety, sadness, and positive emotion in non-MH subreddit posts were significant predictors for posting in MH subreddits. The following subsections present the detailed results for the three research questions.

### RQ 1: What proportion of reddit users with a BD diagnosis posts in MH and non-MH subreddits?

Two thirds of the 19,685 identified Reddit users who disclosed a BD diagnosis posted in both MH and non-MH subreddits (Fig 2). One quarter never posted in MH subreddits and only 6.8% posted exclusively in MH subreddits. The majority (54.7%) of users in the S-BiDD dataset were active contributors in both MH and non-MH subreddits. Most users with less than eight

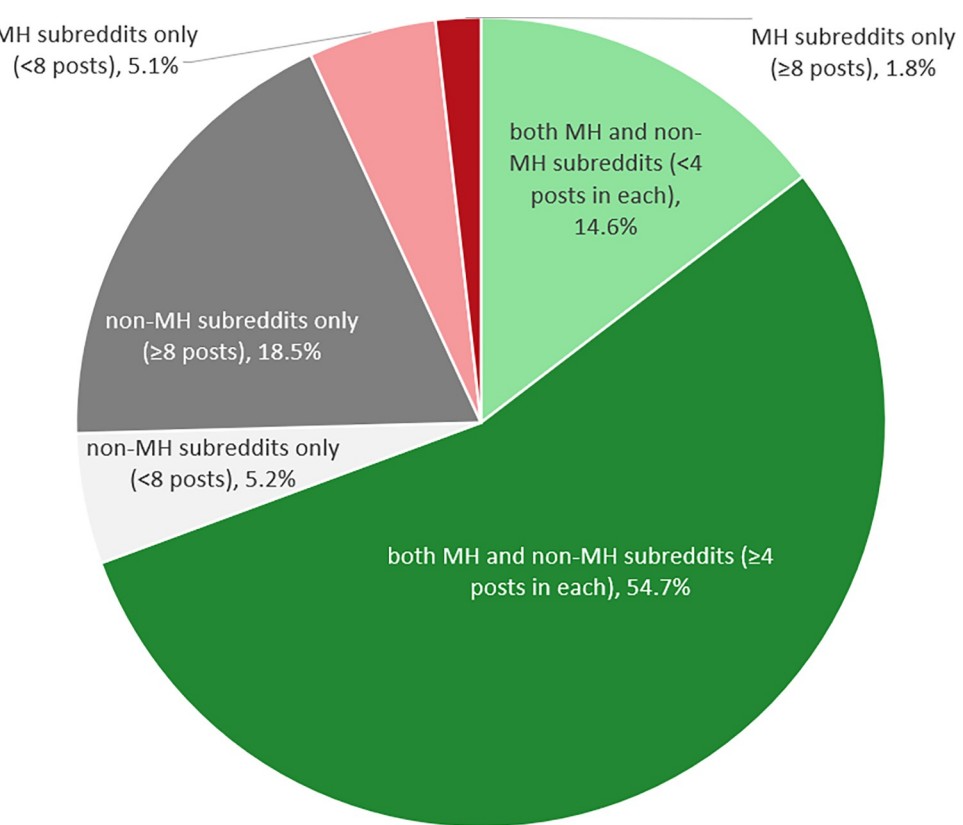

**Fig 2. Reddit users with a self-reported BD diagnosis (n = 19,685) according to the subreddit types in which they post.**

posts posted in both MH and non-MH subreddits (14.6%) and equally likely posted only in MH/non-MH subreddits (5.1%/5.2%).

### RQ 2: What differences exist in the emotions that reddit users with a BD diagnosis express in MH and non-MH subreddit posts?

N = 9,821 users (49.9% of total) with 6,493,626 posts of at least 25 words (11.4% in MH subreddits) met the criteria set out in the 'Reddit user identification and inclusion and exclusion criteria' section as active contributors in both MH and non-MH subreddits with at least four posts in each. There is a significant difference between users' posts in MH and non-MH subreddits for each preselected LIWC variable (see Table 2). Users exhibited slightly less positive emotion in their MH subreddit posts but more anxiety and sadness, and they used much more first-person singular pronouns.

**Table 2. Within user differences between MH and non-MH subreddit posts.**

|  | MH posts | | Non-MH posts | | Dependent t-test, *** = p < 0.001 with Bonferroni correction for 5 comparisons | | |
|---|---|---|---|---|---|---|---|
|  | mean | std | mean | std | $P$ | effect size (Cohen's d) | effect size interpretation [103] |
| positive emotion | 3.48 | 1.08 | 3.64 | 0.93 | < .001 | -0.20 | Small |
| anxiety | 0.67 | 0.42 | 0.37 | 0.22 | < .001 | 0.97 | Large |
| anger | 0.75 | 0.49 | 0.78 | 0.44 | < .001 | -0.07 | Very small |
| sadness | 0.81 | 0.44 | 0.45 | 0.24 | < .001 | 1.11 | Large |
| 1st pers. sg. | 8.82 | 2.28 | 6.39 | 1.96 | < .001 | 1.63 | Very large |

**Table 3. Model fit comparison for controls only and controls with LIWC variables.**

| Model name | Nested / simpler Model | Predictors added | Model fit | | | | LRT against nested | |
|---|---|---|---|---|---|---|---|---|
| | | | AIC | BIC | LL | df | df | $X^2$ |
| Controls | - | age, gender, active days, activity | 10464 | 10500 | -5227 | 10153 | | |
| Controls + LIWC | Controls | posemo, anxiety, anger, sadness, 1$^{st}$ pers. sg. | 10403 | 10476 | -5192 | 10148 | 5 | 70.12, $P < .001$ |

N subjects = 10,309; table key: LRT: likelihood ratio test, AIC: Akaike Information Criterion, BIC: Bayesian Information Criterion, LL: log likelihood, df: degrees of freedom, $X^2$: Analysis of deviance (deviance $X^2$, $P$ ($>X^2$))

## RQ 3: How does mood differ between users who post in MH subreddits and those who only post in non-MH subreddits?

N = 10,158 users (51.6% of total) met the criteria stated in the 'Reddit user identification and inclusion and exclusion criteria' section and were included in the regression model. Of these 2,312 (22.8%) had only posted in non-MH subreddits. S2 Table provides descriptive statistics of the dependent variables. Logistic regression assumptions were checked according to Field et al. [104] (see S3 Appendix for details). The regression model including the LIWC variables as predictors fit the data better than the model with the control variables only (see Table 3).

According to the regression results for the full model in Table 4, Reddit users with feminine gender are 33% more likely to post in MH subreddits compared to users with masculine gender (see S4 Appendix for how this was calculated from the odds ratios of the coefficients). Moreover, higher levels of mean positive emotions, anxiety, and sadness were associated with increased odds to post in MH subreddits. Although the timespan for how long users were active on Reddit was a significant predictor, the coefficient is very small, thus practically negligible. Users' age, their average posts per day, anger, and first person singular pronoun use were not significantly related to whether someone posted in MH subreddits.

**Table 4. Results for the glm regression model including controls and LIWC variables.**

| Model Controls + LIWC | Est/Beta | SE | 95% CI | z | P | Odds ratio (exp(coeff)) | |
|---|---|---|---|---|---|---|---|
| | | | | | | Est/Beta | 95% CI |
| (Intercept) | 0.56 | 0.17 | 0.22–0.90 | 3.21 | 0.001 | 1.75 | 1.24–2.47 |
| age | < .001 | < .001 | -0.01 – < .001 | -0.24 | 0.81 | 1.00 | 0.99–1.00 |
| gender | -0.79 | 0.06 | -0.90 – -0.68 | -14.33 | < .001 | 0.45 | 0.41–0.51 |
| active days | < .001 | < .001 | < .001– < .001 | 11.18 | < .001 | 1.00 | 1.00–1.00 |
| activity | -0.01 | 0.01 | -0.02–0.01 | -1.17 | 0.24 | 0.99 | 0.98–1.01 |
| posemo | 0.05 | 0.01 | 0.03–0.08 | 4.94 | < .001 | 1.06 | 1.03–1.08 |
| anxiety | 0.62 | 0.15 | 0.32–0.91 | 4.08 | < .001 | 1.85 | 1.39–2.51 |
| anger | -0.09 | 0.04 | -0.17 – -0.01 | -2.10 | 0.04 | 0.92 | 0.85–0.99 |
| sadness | 0.41 | 0.12 | 0.18–0.63 | 3.49 | < .001 | 1.50 | 1.20–1.89 |
| 1$^{st}$ pers. sg. | 0.02 | 0.02 | -0.02–0.05 | 0.97 | 0.33 | 1.02 | 0.98–1.05 |
| | | | Model fit | | | | |
| Pseudo $R^2$ | | Hosmer and Lemeshow | | | | Cox and Snell | Nagelkerke |
| | | 0.047 | | | | 0.049 | 0.075 |

Key: p-values for fixed effects calculated using Satterthwaites approximations.

Confidence Intervals have been calculated using the Wald method.

Model equation: glm(posted_in_MH ~ age + gender + active days + activity + posemo + anxiety + anger + sadness + 1$^{st}$ pers. sg., family = binomial(link = "logit"))

### Feedback from people with lived experience

Two of the four people with lived experience who commented on the study design agreed to provide feedback on the results. Overall, the results resonated with their experience of using online support forums, particularly the finding that people who experienced more anxiety and sadness were more likely to post in these forums. One of them shared that initially she had the expectation that online support forums would portray a holistic picture of living with BD. However, over time she noticed a strong focus on difficult experiences in the posts while users who posted that they were doing well got almost no reactions. Both volunteers agreed that it might be beneficial to encourage more discussions around living well with BD and aspects that can make life meaningful to balance out the strong focus on symptoms and diagnosis that they perceived in the forums. Neither of them reported concerns about potential for misuse of the research methods or misinterpretation of the results in a manner that could be harmful for people living with BD.

## Discussion

### Strengths of this study

This study shed more light on the posting behaviour of Reddit users with a self-reported BD diagnosis and provides the first quantitative evidence on the relationship between mood and posting in BD support forums. A strength of this study was The automatic user identification via self-reported diagnosis statements, which resulted in a large sample of participants. Since the data collection did not require direct interaction with the social media users, there was no researcher influence on the behaviour of the participants. Moreover, the psychological text analysis tool LIWC provided easily interpretable measures for in-the-moment emotion assessments, unbiased by self-report. Its wide use facilitated comparison with related work and its psychometric validity to measure emotions has been well established [71]. However, due to a lack of previous research, the study was necessarily exploratory. Future research should be carried out to confirm the findings and explore possible causal relationships between posting behaviour and mood.

### RQ1: What proportion of reddit users with a BD diagnosis posts in MH and non-MH subreddits?

RQ 1 results showed that most (> 50%) Reddit users with a BD diagnosis frequently post both in MH and non-MH subreddits. This exceeds the one third of Reddit users who had posted at least 50 words in both anxiety and non-anxiety-related subreddits identified by Ireland and Iserman [40].

A specific feature of Reddit are throwaway accounts. Since Reddit accounts only require a username for registration, users may create single-use online identities to discuss sensitive topics such as MH issues even more anonymously than via the main account through which they usually engage on the platform [27]. Separating an online identity from one's offline identity or a more holistic online identity (the main Reddit account) may be a main factor of online disinhibition [105, 106], causing people to act differently than in identifiable online settings [107]. De Choudhury and De [27] found that 4.5% of the 1,209 Reddit users who had posted in one of eight MH subreddits in their dataset used a throwaway account as identified by usernames matching *throw*; 61% of these user accounts had made exactly one post. Similarly, in the present study, 3.9% of users had made exactly one post. Moreover, user accounts with less than eight posts equally likely posted only in MH or non-MH subreddits. This suggests that single-use accounts are not necessarily favoured for posting about MH issues.

The fact that so few Reddit users with a self-reported BD diagnosis seem to make use of throwaway accounts and (by inference) to compartmentalise their Reddit identity into only MH vs. non-MH-related concerns is encouraging for further research on the S-BiDD dataset. While [40] considered users with anxiety concerns who post in both MH and non-MH subreddits to be atypical, this does not seem to be the case for users who disclose a BD diagnosis. The present findings imply that most user profiles with a self-disclosed BD diagnosis can be identified as users' main profiles: these profiles are associated with posts covering several aspects of their interests and concerns, which may more closely reflect them compared to throwaway accounts.

The RQ 1 findings also demonstrate that the eligibility criteria for the subsequent RQs that require users to have at least four posts in MH and non-MH subreddits are not too restrictive but capture more than half of the Reddit users who disclose a BD diagnosis.

## RQ 2: What differences exist in the emotions that Reddit users with a BD diagnosis express in MH and non-MH subreddit posts?

The comparison between users' posts in MH and non-MH subreddit posts showed that, in MH subreddits, users expressed somewhat fewer positive emotions, more anxiety and sadness, and used much more first-person singular pronouns. These findings aligned with the expectations of people with lived experience that people may mainly post in MH support forums about MH struggles and when experiencing mental distress. They also aligned with the finding from Ireland and Iserman [40] that people used more anxious language in anxiety-focused subreddits compared to their posts in other subreddits. Increased use of first person singular pronouns has been mainly linked to mental distress [84], although positive I-talk may also occur [84].

The results demonstrate a higher presence of negative emotions and self-focus in MH subreddit posts, which previous research links to rumination and low mood [108, 109]. However, this analysis cannot determine why these differences occur. Possible explanations that future research needs to test are that Reddit users in a lower mood state may be more likely to post in MH subreddits or that they may feel more comfortable expressing negative emotions there.

## RQ 3: How does mood differ between users who post in MH subreddits and those who only post in non-MH subreddits?

Users who exhibited more anxiety and sadness in their non-MH subreddit posts, behaviours that constitute core symptoms of low mood [102], were more likely to post in MH subreddits. This was expected by people with lived experience who reported that they mainly used online support forums when experiencing low mood. Interestingly, exhibiting more positive emotions also increased the odds of posting in MH subreddits, although the effect was an order of magnitude smaller than for anxiety and sadness. Taken together, these findings could indicate that users who experienced more extreme mood, hence stronger BD symptoms, are more likely to post in MH subreddits.

Another finding was that users with a feminine gender identity were 33% more likely to post in MH subreddits compared to users with a masculine gender identity. While biological men and women are equally likely to meet BD criteria [2, 5], Jagfeld et al. [39] already established that slightly more than half of the Reddit users who self-report a BD diagnosis had a feminine gender, although two thirds of Reddit users in general are male [110]. Of the users who never posted in MH subreddits, 60.5% had a masculine gender, while 59.2% of the users who posted in MH subreddits had a feminine gender. This proportion is in line with an average of 58% of women enrolled in 55 trials included in a large meta-analysis of psychological

interventions for BD [111]. Overall, the finding that users with a feminine gender identity are more likely to post in MH forums is in line with previous research that found men were less likely to disclose and seek support for MH issues [94–96].

Also using LIWC, De Choudhury et al. [83] found that feminine-identified users who disclosed MH issues on Twitter exhibit more anxiety and sadness in their posts and less positive emotions compared to masculine-identified users who disclosed MH issues. This mirrors our findings on the differences between Reddit users who post or do not post in MH subreddits. To check whether our findings of mood differences between Reddit users who post versus do not post in MH subreddits could be attributed to the gender imbalance in the outcome groups, an additional regression model was evaluated with gender-balanced samples for both outcome groups. However, the results for the gender-balanced sample of 8,092 users did not change substantially compared to the original sample (see S5 Appendix).

## Implications

**Implications for research.**   In general, a better understanding of people with a BD diagnosis who post on Reddit is important to contextualise other research on this kind of data, with Reddit being one of the most frequently used data sources for research on BD using NLP methods [41]. More specifically, the findings for RQ 2 demonstrated substantial differences in LIWC scores between posts in MH and non-MH subreddits. Although expected, this had not been empirically tested so far. Providing quantitative evidence for this is important because what seems 'common sense' is actually only so when it has been empirically demonstrated [112]. The RQ2 findings provide strong support for all studies relying on emotion text analysis, specifically comparative language analyses and detection systems of people with and without MH issues, to only consider posts in a comparable context as done in more recent research [44–46], compared to earlier work [42, 43]. For studies that consider recruiting participants from online support forums, it is relevant to know that mainly users who may experience stronger mood symptoms tend to be active there. There is also a higher proportion of feminine-identified users who post in these forums, which is similar to the proportion of women recruited into clinical trials. Relying only on online support forums for recruitment may thus introduce sampling biases.

**Implications for practice.**   In general, this study provides information to MH professionals about online support forums for BD–something for which international BD experts highlighted a currently critical need [26]. Since the analyses aggregated several thousands of Reddit users and found mostly small to medium effect sizes, the findings are relevant on the group level and do not permit conclusions for individual people. It may be helpful to inform people living with BD who (want to) use online support forums that people living well with MH issues might be underrepresented there and recommend additional information sources to obtain a more balanced picture of living with BD. Forum providers and moderators might consider encouraging more discussions around living well with BD to potentially balance some of the content exhibiting higher levels of anxiety and sadness. On the other hand, online support forums may be a useful outlet where people living with BD feel safe and welcome to express their more intense emotions, particularly anxiety and sadness. Spaces where people can express intense negative emotions are important because they counteract suppression and avoidance of such emotions. These are considered maladaptive emotion-regulation strategies and have been linked to depression and anxiety [113].

## Limitations and future work

This study has three limitations that future research could address. It was not possible to check the veracity of self-reported diagnoses, although it appears unlikely that people would

deliberately unfaithfully identify with a stigmatised MH condition. Cluss et al. [114] verified that 93% of 804 individuals who self-reported BD diagnosis met criteria for a lifetime BD diagnosis according to an in-person structured diagnostic interview. Nevertheless, future complementary research could conduct diagnostic interviews to determine the eligibility of participants, which would also allow inclusion of people who meet criteria for BD but have not received a diagnosis yet. LIWC is only one tool to study the mood of social media users. Future work could triangulate the present results with emotion text analysis systems trained via machine learning that may be more accurate in context but less interpretable than the dictionary-based LIWC tool [82], as well as with self-reported questionnaires and qualitative interviews. Finally, his study only considered data until March 2019. While online MH forum usage has likely increased since then, fuelled by the global Covid-19 pandemic [115], there are no apparent reasons why the usage patterns found here would have changed. However, future research could replicate these analyses with more recent data and consider changes over time. Future research should also explore whether the present findings are specific to BD or mood or MH issues more general.

Additionally, future research could extend the present findings by looking at what effects engaging with online MH support forums has on users' mood as well as on person-centred holistic wellbeing in longitudinal studies. Recommendations for people living with BD for how to use social media and online support forums to achieve good long-term quality of life [116] or personal recovery [117] was something that people with lived experience of BD considered as particularly helpful to develop. The ongoing Improving Peer Online Forums study, which aims to find out how online MH forums work, why some work better than others, and why some people find them helpful and others do not, may yield promising evidence for this [118].

## Conclusions

This exploratory study was the first quantitative investigation to provide information on forum posting behaviour and its relationship with emotions and mood in Reddit users with a self-reported BD diagnosis. The study had three main findings: First, Reddit users with a self-reported BD diagnosis frequently post in both MH and non-MH subreddits. Second, for the same users, there are large differences in the expressed emotions in MH versus non-MH subreddit posts. Third, feminine gender, higher anxiety, sadness and positive emotions were associated with significantly increased odds that a Reddit user posted to MH subreddits. These findings have important implications for research on Reddit and online MH forums in general, as well as for the design of online MH forums, and for recommendations concerning their use to people living with BD.

## Supporting information

**S1 Table. Top 10 subreddits with most posts in the dataset for subreddits in the BD or MH subreddit list (excl. BD) and subreddits not in the MH subreddit list.** The number of non-MH subreddits is the number of unique subreddits in which users in the dataset posted that are not in the MH subreddit list. The users in the dataset only posted in 36 of the 37 pre-identified BD-specific subreddits and in 116 of the 121 (158–37) not BD-specific subreddits in the MH subreddit list.
(DOCX)

**S2 Table. Descriptive statistics and between user differences for users in the regression model.**
(DOCX)

**S1 Appendix. Details on the creation of the BD and MH subreddit lists.**
(DOCX)

**S2 Appendix. Method details for the logistic regression model with controls and LIWC variables.**
(DOCX)

**S3 Appendix. Assumptions checking for the logistic regression model with controls and LIWC variables.**
(DOCX)

**S4 Appendix. Probability calculation for the gender coefficient.**
(DOCX)

**S5 Appendix. Regression results for subsampled dataset with gender-balanced outcome groups.**
(DOCX)

## Acknowledgments

The authors would like to thank the people with lived experience of bipolar disorder, who provided input to the design of this study and the results and their implications. We would also like to thank Enrica Troiano for insightful discussions on emotion text analysis. We are grateful to the UCREL research group at Lancaster University for multiple opportunities to discuss and present this work, with particular thanks to Andrew Moore and Daisy Harvey for helpful comments and Matthew Coole for testing the code release.

## Author Contributions

**Conceptualization:** Glorianna Jagfeld, Fiona Lobban, Paul Rayson, Steven Jones.

**Data curation:** Glorianna Jagfeld.

**Formal analysis:** Glorianna Jagfeld.

**Funding acquisition:** Fiona Lobban, Paul Rayson, Steven Jones.

**Investigation:** Glorianna Jagfeld.

**Methodology:** Glorianna Jagfeld, Fiona Lobban, Robert Davies, Ryan L. Boyd, Paul Rayson, Steven Jones.

**Project administration:** Glorianna Jagfeld.

**Resources:** Glorianna Jagfeld, Ryan L. Boyd.

**Software:** Glorianna Jagfeld.

**Supervision:** Fiona Lobban, Paul Rayson, Steven Jones.

**Validation:** Glorianna Jagfeld.

**Visualization:** Glorianna Jagfeld.

**Writing – original draft:** Glorianna Jagfeld.

**Writing – review & editing:** Glorianna Jagfeld, Fiona Lobban, Robert Davies, Ryan L. Boyd, Paul Rayson, Steven Jones.

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
