## [Decision Letter · Decision Letter 0]

14 Nov 2022

PONE-D-22-26677Posting patterns in peer online support forums and their associations with emotions and mood in bipolar disorder: exploratory analysisPLOS ONE

Dear Dr. Jagfeld,

Thank you for submitting your manuscript to PLOS ONE. After careful consideration, we feel that it has merit but does not fully meet PLOS ONE’s publication criteria as it currently stands. Therefore, we invite you to submit a revised version of the manuscript that addresses the points raised during the review process.

We look forward to receiving your revised manuscript.

Kind regards,

Yaakov HaCohen-Kerner, Ph.D.

Academic Editor

PLOS ONE

Journal Requirements:

3. Your ethics statement should only appear in the Methods section of your manuscript. If your ethics statement is written in any section besides the Methods, please move it to the Methods section and delete it from any other section. Please ensure that your ethics statement is included in your manuscript, as the ethics statement entered into the online submission form will not be published alongside your manuscri

Additional Editor Comments:

Dear authors,

Your study deal with a very important subject.

Your paper contains many well-written components (e.g., the introduction chapter, research questions, results, and analysis).

However, both reviewers recommend major revisions.

They recommend using additional and more suitable tools for your research.

They wrote that various points in your paper are not clear and not elaborated enough (many details were given by Reviewer 1).

According to Reviewer 2, your study does not provide either NLP contributions or state-of-the-art results.

Both reviewers also report various types of minor corrections.

Therefore, unfortunately, the paper in its current state requires too many revisions to be accepted.

I wish you good luck and success in your future studies and papers including this paper.

Yaakov

Reviewers' comments:

Reviewer's Responses to Questions

**Comments to the Author**

1. Is the manuscript technically sound, and do the data support the conclusions?

Reviewer #1: Yes

Reviewer #2: Yes

2. Has the statistical analysis been performed appropriately and rigorously? 

Reviewer #1: N/A

Reviewer #2: Yes

3. Have the authors made all data underlying the findings in their manuscript fully available?

Reviewer #1: Yes

Reviewer #2: Yes

4. Is the manuscript presented in an intelligible fashion and written in standard English?

Reviewer #1: Yes

Reviewer #2: Yes

5. Review Comments to the Author

Reviewer #1: This study employs psychological text analysis techniques in online forums posts (Reddit posts) to explore

posting patterns and their relationships with emotions and mood of people with a Bipolar Disorder (BD)

diagnosis.

In particular, authors investigated and answered the following research questions:

RQ 1: What proportion of Reddit users with a BD diagnosis posts in MH and non-MH subreddits?

RQ 2: What differences exist in the emotions that Reddit users with a BD diagnosis express in MH and

non-MH subreddit posts?

RQ 3: How does mood differ between Reddit users with a BD diagnosis who post in MH subreddits and

those who only post in non-MH subreddits?

Contribution:

User identification and dataset construction in Reddit posts

Study of relationships between posting patterns and people’s emotions

Positive:

A complete introduction is given

Research questions are presented and addressed very well

Paper is well written and easy to follow

Sufficient details are given and claims are justified with proper citations

The results and analysis are tabulated and discussed in detail

Negative:

The body text alignment need correction

Use [] for citations instead of ()

Figure 1 is not referred in text body

What are the main contributions of your work?

The use of LR model is not presented well, how did you model the task?

Improvements:

The use of LR model to address RQ3 could be elaborated more. From a NLP perspective it

requires to describe how it was trained, what statistics of dataset was used and more detail about

the procedure.

The contributions of the article could be listed

More graphical charts and figure would help for a better representation of your work

Since you have explored emotion, you may consider exploring Symanto API as well for future

work, where it will provides you with tool for extract if the text is rational or emotional.

Sentiments Analysis, Emotion detection, and psychological tools are other interesting and related

tools for you.

Reviewer #2: The study makes no contribution to the field of natural language processing. For emotion text analysis, the dictionary-based LIWC method does not provide state-of-the-art NLP results. Using more relevant tools may provide different outcomes.

I am not an expert in mental health, but the research result appears to be quite straightforward.

The author covers and refers to prior work in a methodical manner, and the statistical analysis appears to be valid. I believe that the author should have compared user behavior over several time intervals. In particular, How was the user's depression represented in MH and non-MH postings on a particular day or week when they experienced it?

Minors:

Line: 264, 267: Error! Reference source not found

Line 291: formatting issue

6. PLOS authors have the option to publish the peer review history of their article (what does this mean?). If published, this will include your full peer review and any attached files.

Reviewer #1: **Yes: **Grigori Sidorov

Reviewer #2: No

---

## [Author Response · Author response to Decision Letter 0]

3 Jan 2023

Please refer to our detailed reviewer response in the submitted file BipolarPostingMood_PLOS_R1_ReviewerResponse.pdf.

---

## [Decision Letter · Decision Letter 1]

10 May 2023

PONE-D-22-26677R1Posting patterns in peer online support forums and their associations with emotions and mood in bipolar disorder: Exploratory analysisPLOS ONE

Dear Dr. Jagfeld,

Thank you for submitting your manuscript to PLOS ONE. After careful consideration, we feel that it has merit but does not fully meet PLOS ONE’s publication criteria as it currently stands. Therefore, we invite you to submit a revised version of the manuscript that addresses the points raised during the review process.

We look forward to receiving your revised manuscript.

Kind regards,

Michele Fornaro

Academic Editor

PLOS ONE

Reviewers' comments:

Reviewer's Responses to Questions

**Comments to the Author**

1. If the authors have adequately addressed your comments raised in a previous round of review and you feel that this manuscript is now acceptable for publication, you may indicate that here to bypass the “Comments to the Author” section, enter your conflict of interest statement in the “Confidential to Editor” section, and submit your "Accept" recommendation.

Reviewer #3: All comments have been addressed

Reviewer #4: (No Response)

2. Is the manuscript technically sound, and do the data support the conclusions?

Reviewer #3: Yes

Reviewer #4: Yes

3. Has the statistical analysis been performed appropriately and rigorously? 

Reviewer #3: I Don't Know

Reviewer #4: Yes

4. Have the authors made all data underlying the findings in their manuscript fully available?

Reviewer #3: Yes

Reviewer #4: Yes

5. Is the manuscript presented in an intelligible fashion and written in standard English?

Reviewer #3: No

Reviewer #4: Yes

6. Review Comments to the Author

Reviewer #3: This is an interesting study. Bringing in lived experience was a nice touch. I do, however, have a few suggestions. One is that people may seek out forums like Reddit because they feel more comfortable even if they do have access to health care. I would urge you to go over the introduction again because it seems to be redundant in places, and I found the description of the information in Table 3 difficult to understand. Your readers might not always be sophisticated statistically, particularly in the case of mental health. Finally, it would be great if you had some ideas about how to help participants cope better.

Reviewer #4: 1. The methods are hard to read. The presence of subsections is confusing and does not allow prompt overall comprehension. In addition, the inclusion and exclusion criteria are not easily accessible; some are even in the results section. I suggest the authors rewrite this section considering a different sub-section scheme: i) inclusion and exclusion criteria (including self-report diagnosis, comorbidities, number of posts, karma, user age when posting, etc.); ii) Reddit user identification, mental health subreddit identification (also, the ‘line 165’ includes results (quantitative data) that should be moved to the appropriate section); iii) language analysis (well-written btw); iv) outcomes (research questions); v) ethical considerations; vi) involvement of people with lived experience (herein some sentences (e.g., lines 284-291), are results (even if narrative) and should not be presented in methods.

Also, there is no indication of employed data analysis (e.g., cohen’s d, OR, chi-square). They should be stated and added as a sub-section of methods.

The sequence of the sub-sections is indicative and not mandatory at all.

I do not really like the RQ presentation. I would rephrase them in ‘outcomes’ to be presented in a separate subsection and move the scattered inclusion criteria to a different section.

How did you handle the [deleted] users, given that you included accounts with <4 posts? Did the program automatically exclude them?

2. Line 236: ‘Some control variables’: what are we talking about? Is this presented in methods?

3. Line 239: why not >4 posts like before? By the way, this should be moved in inclusion criteria in methods.

4. Line 307: no, you did not write the eligibility criteria..

5. Table 2: anger effect size interpretation is ‘no effect’, but the p-value is <.001, please check all p-values.

6. Table 4: active days. 95%C.I. <.001, <.001, as well as the p-value. Please double-check.

7. Results: Please consider moving line 165 at the start of the results, along with the results summary. Before the sub-sections, you can write Fig. 2 provides a visual summary of all results. Also, the RQ style could be rephrased to get outcome-like headings.

A style suggestion: when you state ‘Fig.1 shows, table 2 shows’, you may rephrase the sentence and put (Table 1) or (Fig.2) at the end.

8. Discussion: lines 487-490 should be the first sentences of the discussion. Remove the ‘principal results’ heading and include lines 358-360 to 487-490. Lines 437-438 are redundant. Lines 451-453 should be discussed earlier because they talk about a general result.

You can make this section more reader-friendly by removing the RQ subsections and trying to discuss the results by dividing the outcomes with a simple carriage return. Lines 442-449 are a staccato of the RQ2 discussion (earlier stated). The ‘implications for practice’ sub-section is fine. The section ‘strengths, limitations, and future work’ should be changed to ‘limitations of the study’, and the ‘strengths of the study’ should be presented at the start of the discussion. Try avoiding stating first, second, and third.

9. Abbreviations: it is redundant to present abbreviations in a separate section. You can write the (abbreviation) the first time you use a phrase, then write the abbreviation. Watch out, MH is already present at line 68, while ‘mental health’ is stated at line 72 for the first time. Double-check all other abbreviations.

10. Miscellaneous: Line 61: ‘extreme high and low mood states’ is not really a good way to describe the core criteria of bipolar disorder diagnosis. Consider rephrasing it.

Lines 63-64 content is redundant (you state the countries in brackets, then state them again).

Lines 89-91 (until [35]) should go before line 83, and the two paragraphs should merge and harmonize.

Line 130: what does ‘emotional tone improves’ mean? I struggle to believe that own’s emotional tone improves while talking in the suicidality subreddit.

Line 164: bipolar disorder automatically excludes the possibility of a ‘depression’ diagnosis (you talk about DSM-5, then write depression) or, better, a major depressive disorder. The only chance is a major depressive episode in a bipolar patient. Please rephrase.

7. PLOS authors have the option to publish the peer review history of their article (what does this mean?). If published, this will include your full peer review and any attached files.

Reviewer #3: No

Reviewer #4: No

---

## [Author Response · Author response to Decision Letter 1]

16 Jun 2023

Please refer to our detailed responses in the file BipolarPostingMood_PLOS_R2_ReviewerResponse.docx.

---

## [Decision Letter · Decision Letter 2]

29 Aug 2023

Posting patterns in peer online support forums and their associations with emotions and mood in bipolar disorder: Exploratory analysis

PONE-D-22-26677R2

Dear Dr. Jagfeld,

We’re pleased to inform you that your manuscript has been judged scientifically suitable for publication and will be formally accepted for publication once it meets all outstanding technical requirements.

Kind regards,

Michele Fornaro

Academic Editor

PLOS ONE
---

## [Editor Report · Acceptance letter]

15 Sep 2023

PONE-D-22-26677R2 

Posting patterns in peer online support forums and their associations with emotions and mood in bipolar disorder: Exploratory analysis 

Dear Dr. Jagfeld:

I'm pleased to inform you that your manuscript has been deemed suitable for publication in PLOS ONE. Congratulations! Your manuscript is now with our production department. 

Kind regards, 

on behalf of

Dr. Michele Fornaro 

Academic Editor

PLOS ONE